# Mitral Valve Prolapse, Arrhythmias, and Sudden Cardiac Death: The Role of Multimodality Imaging to Detect High-Risk Features

**DOI:** 10.3390/diagnostics11040683

**Published:** 2021-04-10

**Authors:** Anna Giulia Pavon, Pierre Monney, Juerg Schwitter

**Affiliations:** 1Cardiac MR Center (CRMC), Lausanne University Hospital (CHUV), 1100 Lausanne, Switzerland; Pierre.Monney@chuv.ch (P.M.); jurg.schwitter@chuv.ch (J.S.); 2Cardiovascular Department, Division of Cardiology, Lausanne University Hospital (CHUV), 1100 Lausanne, Switzerland; 3Faculty of Biology and Medicine, University of Lausanne (UniL), 1100 Lausanne, Switzerland

**Keywords:** mitral valve prolapse, arrhythmias, cardiovascular magnetic resonance

## Abstract

Mitral valve prolapse (MVP) was first described in the 1960s, and it is usually a benign condition. However, a subtype of patients are known to have a higher incidence of ventricular arrhythmias and sudden cardiac death, the so called “arrhythmic MVP.” In recent years, several studies have been published to identify the most important clinical features to distinguish the benign form from the potentially lethal one in order to personalize patient’s treatment and follow-up. In this review, we specifically focused on *red flags* for increased arrhythmic risk to whom the cardiologist must be aware of while performing a cardiovascular imaging evaluation in patients with MVP.

## 1. Mitral Valve and Arrhythmias: A Long Story Short

In the recent years, the scientific community has begun to pay increasing attention to mitral valve prolapse (MVP) due to its correlation with ventricular arrhythmias (VAs) and an increased risk of sudden cardiac death (SCD). The prolapsing mitral valve was first described in the late 1960s [1,2], but it is only recently that the term “arrhythmic MVP” (aMVP) has been introduced [3]. The syndrome is characterized by complex premature ventricular beats (PVBs) arising from one or both papillary muscles (PMs) and an increased risk of SCD, estimated to be between 16 and 41 per 10,000 per year (0.2%–0.4%), compared to the general population [4,5]. Several studies have identified individual risk factors associated with VA, including (1) female sex [3], (2) bileaflet prolapse with marked leaflet redundancy, (3) mitral annulus abnormalities [6], (4) electrocardiographic repolarization abnormalities specifically in the inferolateral leads [7], (5) frequent and complex PVB [8], and (6) presence of myocardial fibrosis at the level of the PMs [3]. While the connection between MVP and complex VA is well established, until now, no specific recommendations have yet been proposed to guide clinical management and to estimate the arrhythmic risk in MVP patients. We here summarize the most important morphological features to recognize during cardiac imaging of MVP patients, as they may serve as *red flags* for increased arrhythmic risk.

## 2. Imaging the MVP: What Are the “High-Risk” Features to Look For?

Cardiac imaging plays a key role in the diagnosis of MVP and in the detection of the major arrhythmic high-risk features. Overall, the reported incidence of premature ventricular beats (PVBs) in MVP may vary from 49% to 85% in the adult population. Typically, the most common site of origin of these PVBs is the postero-basal portion of the Left Ventricle (LV) at the level of PM, or they are of fascicular origin [9,10,11]. This feature is consistent with the hypothesis that the mechanical irritation produced by the prolapsing valve could be the trigger of these arrhythmias. Transthoracic echocardiography (TTE) is the first-line test to diagnose MVP and it may detect several morphological high-risk features. However, when MVP is suspected, advanced cardiac magnetic resonance imaging (CMR) may improve the collection of high-risk features.

### 2.1. Leaflets and Mitral Apparatus

TTE and transesophageal echocardiography (TEE) are the reference techniques to evaluate leaflet anatomy [12]. The “classical” definition of MVP is a single or bileaflet prolapse of at least 2 mm beyond the long-axis mitral annular plane, with or without leaflet thickening. MVP should only be diagnosed from a strict long-axis view (i.e., a parasternal or apical long-axis view), as these views cut the saddle-shaped mitral annulus at its highest point: Looking for MVP from other echocardiographic views may result in the overdiagnosis if MVP due to the complex three-dimensional shape of the mitral annulus [13]. Limitations due to a poor acoustic window may be easily overcome by TEE, and the use of three-dimensional echocardiography additionally provides a detailed evaluation of the mitral valve anatomy in “surgical view,” of the mechanism of regurgitation (including the distinction between Barlow’s disease and fibroelastic deficiency), and of the surgical reparability of the valve. While the degree of mitral regurgitation itself has not been associated with VA or SCD [7,8,14], the combination of a myxomatous valve (Barlow’s disease) with bileaflet involvement and marked leaflet redundancy during echocardiography should already raise the suspicion of aMVP and justify dedicated investigation to look for additional high-risk features associated with the MVP syndrome [7]. The tugging of the posteromedial PM in mid-systole by the myxomatous prolapsing leaflets causes the adjacent postero-basal left ventricular wall to be pulled sharply toward the apex. This brisk mid-systolic apical motion can be captured with tissue Doppler imaging of the postero-basal segment in the long-axis apical view, showing a spiked configuration of the Doppler spectrum in mid-systole, the so-called “Pickelhaube” sign (Figure 1B) [15]. It reflects the abrupt stretching of the myocardium and papillary muscles by the prolapsing valve, which may serve as a mechanical trigger of Premature Ventricular Contraction (PVC) or ventricular arrhythmia. Recent data have suggested that a spiked tissue Doppler systolic velocity ≥16 cm/s is a risk marker and should be included in the routine echocardiographic evaluation of MVP patients [15]. Regional systolic myocardial motion can also be captured by speckle-tracking echocardiography, and in a cohort of MVP patients with significant mitral regurgitation, lower global longitudinal strain values and prolonged mechanical dispersion are independently associated with the occurrence of symptomatic VA [16,17]. Pathological deformation patterns with increased pre-stretch and post-systolic contraction have also been described in the basal segments of MVP patients [18], and patients with Barlow’s disease show enhanced strain in the basal segments compared to patients with fibro-elastic deficiency or normal controls [19]. The potential importance of these features for arrhythmic risk stratification will require further investigation.

### 2.2. Mitro-Annular Disjunction (MAD)

First mentioned by Bharati et al. [20], MAD refers to a displacement of the insertion point of the posterior mitral valve leaflet, which accounts for a wide separation between the left atrial wall and the left ventricle. The diagnosis of MAD is usually made in systole by measuring the distance between the posterior leaflet insertion into the left atrial wall and the base of the LV free wall [21]. This should be performed in a parasternal long axis view in echocardiography or in three-chamber view during CMR [22].

The assessment of MAD is of importance, as it has emerged as an independent risk factor for VA in several studies, probably due to its link to the mechanical stretch of the myocardium [3]. While larger degrees of MAD correlate with higher incidence of complex VA, the cut-off for MAD diagnosis is not uniformly defined. Usually the histological reference of >5 mm at the posterior leaflet level, firstly described by Hutchins et al., is considered in most echocardiographic studies (Figure 1A) [23]. However, it must be kept in mind that the circumferential extension of MAD is variable, and sometimes, the identification of MAD using TTE may be challenging [24]. The better acoustic window on TEE may overcome the disadvantage of insufficient TTE image quality, but comes at the cost of a more invasive examination [12]. On the contrary, the extent of longitudinal MAD distance located in the posterolateral wall is easily assessed by CMR [22] (Figure 1C). Essayagh et al. [8] compared TTE and CMR for the detection of MAD and found a low sensitivity (65%) but a high specificity (96%) for TTE. A recent study compared TTE, TEE, and CMR and showed only a moderate agreement between TTE and CMR, while a good agreement was found between TEE and CMR [25]. Even if not recommended in routine evaluation, cardiac computed tomography also allows to measure the length of MAD [26].

Notably, an unusual systolic motion of the posterior mitral ring and the adjacent myocardium may also be found in patients with aMVP (“Curling”) [3,10]. This specific movement has been linked in several series with the presence of MAD and it contributes to the paradoxical increase in the mitral annulus diameter during systole and the relative hypertrophy of the hypermobile postero-basal segments [7]. It may be associated with the development of LV fibrosis, accounting for the excessive mobility of the MV apparatus and systolic stretch of the myocardium closely linked to the valve. For the moment, only a visual non-quantitative assessment is typically performed, while no specific cut-off has been identified (Appendix A: Curling motion during transthoracic echocardiography; Appendix A: Curling motion during CMR; Appendix A).

### 2.3. Macroscopic Fibrosis

The presence of fibrosis at the PM and inferior LV wall level is another key feature of aMVP and has been proposed as an independent risk factor for VA and SCD [27]. CMR is the only imaging technique that allows for a detailed evaluation of myocardial tissue characteristics, but until now, it has not been routinely used in patients with MVP.

Basso et al. [27] demonstrated in 2015, by histopathologic analyses in young SCD victims with MVP and trivial mitral regurgitation, a high prevalence of myocardial fibrosis in the LV infero-basal wall and in the PMs. These findings were confirmed in vivo in a subpopulation of MVP patients with complex ventricular arrhythmias, in whom CMR showed a late gadolinium enhancement (LGE) distribution similar to the histopathologic findings observed in SCD victims. These myocardial fibrotic alterations have been confirmed in other studies [6,7,8,14,28], and are probably linked to the mechanical stretch acting upon the valve and the neighboring LV myocardium and could represent the arrhythmic substrate in patients with MVP. However, identification of LGE at the PM level and neighboring LV walls may be challenging even for CMR experts, and additional non-standard views may be needed to best visualize fibrotic regions in the myocardium (Figure 1F). Undoubtedly, this evidence supports the need for a CMR examination with LGE acquisitions in all patients with suspected aMVP.

### 2.4. Interstitial Fibrosis: Is There a Role for T1 Mapping or Extracellular Volume?

Beyond macroscopic fibrosis, recent papers have highlighted the possible role of interstitial fibrosis measured during CMR T1 mapping and extracellular volume (ECV) measurements [29,30]. Macroscopic fibrosis detection by LGE may be challenging, especially in areas where the myocardium is not completely replaced by fibrotic tissue [31]. According to this, other studies have found much lower rates of LGE, with clear CMR demonstration in only a third of patients [32].

Pradella et al. [33] and Bui H et al. [34] found globally higher native and lower post-contrast T1 myocardial values, suggesting the presence of interstitial fibrosis, but no association with complex VA was found (Figure 1D,F). However, their studies suffered from various limitations, in particular, the lack of a control group (patients with mitral insufficiency but without MVP) and the absence of a multivariate analysis considering the presence of other high-risk features. Moreover, Guglielmo et al. highlighted that patients with MVP have higher native T1 times compared to healthy controls, particularly in the basal and mid-LV inferolateral walls; higher T1 values were not correlated with MR severity, indicating that diffuse fibrosis may not only be a consequence of volume overload [35].

In addition to this, myocardial ECV quantification by T1 mapping has emerged as an accurate tool to detect interstitial fibrosis, and it is used as a quantitative marker in ischemic and non-ischemic cardiomyopathies [36,37]. Most importantly, ECV has been found to be an independent predictor of adverse outcomes in patients with heart valvular disease [38,39]. To date, the presence of higher ECV has been demonstrated only in patients with MVP and dyspnea on exertion or with reduced exercise tolerance, but no data so far support an eventual link with VA and SCD [40]. Importantly, ECV has been found to be statistically higher in MAD patients than in those without MAD, in all the basal myocardial segments, with a significant correlation between ECV of the basal inferior wall and MAD size. This finding emphasizes the potential role of interstitial LV fibrosis in the arrhythmogenic abnormalities frequently associated with MAD [41], suggesting that ECV may provide a more comprehensive assessment of LV remodeling induced by MVP.

In summary, the presence of elevated native T1 values, the reduced post-contrast T1 values, and the higher ECV values reported so far suggest that the fibrotic alterations in patients with MVP may go far beyond the presence of macroscopic LGE. Actually, all these observations support the hypothesis that the presence of diffuse fibrosis may also be a trigger for VA, and that it may represent an initial step preceding the development of macroscopic fibrosis detected by LGE sequences. For these reasons, our recommendation would be to routinely use T1 mapping in combination with LGE in the evaluation of these patients, even if the clinical significance of elevated T1 and its association with an increased risk of SCD or arrhythmia in MPV patients needs further investigation and validation.

### 2.5. Inflammation: Should We Look for It?

Recently, the hypothesis of subclinical myocardial inflammation or ischemia as an additional pro-arrhythmic mechanism in patient with MVP has also been evoked as a potential VA trigger [42]. For this, reason Miller et al. presented a pilot study, suggesting that patients with significant degenerative mitral regurgitation (MR) owing to MVP and a history of ventricular ectopy may also evidence of occult inflammation in addition to myocardial fibrosis.

Actually, in 85% of their small series of patients, tracer uptake at the same level of fibrosis found in LGE was demonstrated with ^18^F-fluoro-deoxy-glucose (FDG) positron emission tomography [43]. Despite the small sample size, there was a trend toward a higher burden of scarring in the patients with complex VA, which exhibited a lower FDG uptake compared to patients with minor VA (3.5% vs. 6.9%). According to these results, the authors suggested that the presence of inflammation at the very early stage and of the disease, with just minor VA, might finally lead to replacement fibrosis and to more complex VA.

However, intriguingly, it must be considered that FDG is a standard tracer for inflammation imaging; thus, it is nonspecific and the question of ischemia is not addressed with this method. Moreover, a different cellular glucose uptake at the level of PMs may also be responsible for these results. Thus, for the moment, only a potential link with inflammation may be postulated. Thus, further studies are needed before considering a routine assessment of inflammation in patients with suspected aMVP.

## 3. Management: How Can We Put All Together?

aMVP is a complex syndrome that exposes patients to a higher risk of VA and SCD [3]. Cardiovascular imaging plays a pivotal role in detecting the disease and helps in risk stratification. Considering the high prevalence of MVP and the benign prognosis of this condition for the majority of the patients, cardiologists should be familiar with those reported high-risk features and should actively look for them during every routine TTE imaging. Patients with suspected aMVP based on symptoms, baseline ECG, or TTE should then be referred for more advanced examinations. TTE is the first-line technique for patients with MVP, but since the severity of mitral regurgitation does not appear to be associated with arrhythmic risk [7], cardiologists should carefully look for MAD [24], detect the “Pickelhaube” sign and measure its systolic velocity [15], and estimate the degree of mechanical dispersion with speckle tracking echocardiography [16,17] to identify patients at higher risk. In the setting of suspected aMVP, CMR is able to accurately measure LV volume and to quantify mitral regurgitation [44], and to more precisely detect and assess the severity of MAD. However, the pivotal role of CMR lies in the non-invasive detection of myocardial focal fibrosis, typically in the papillary muscles and the postero-basal myocardium, using LGE acquisitions [3]. Additional quantification of interstitial fibrosis with T1 mapping and ECV measurements might be able to further refine the risk stratification [44].

Finally, some patients may be candidates for mitral valve surgery because of the severity of mitral regurgitation. It is important to remember that no statistically significant data describe the evolution of ventricular arrhythmia burden after surgery and whether surgical mitral valve repair may reduce the overall arrhythmic risk. In a very small and insufficiently powered series of patients [45], mitral valve surgery did not reduce the burden of ventricular arrhythmias in patients with bileaflet MVP. Based on this preliminary observation and awaiting further data, we might speculate that according to the central role of replacement fibrosis in the genesis of arrhythmias, the persistence of intramyocardial fibrosis after surgery may still represent a substrate for electrical re-entry, which may expose the patient to an ongoing arrhythmic risk.

In conclusion, at the present time, the current guidelines do not provide guidance about risk stratification, treatment, and follow-up in aMVP patients. From the perspective of individualized management of patients with suspected aMVP, the high-risk features detected by cardiac imaging represent important tools to identify the minority of patients for whom a more aggressive management might be considered, including pharmacological treatment, catheter ablation procedures [46], implantable cardioverter defibrillator implantation, or mitral valve surgery [47].

## 4. Mitral Valve Prolapse in Real Life: When Prevention Makes the Difference


***Case 1 (*Figure 2*): Primary prevention ICD implantation after multi-modality imaging work-up***


A 22 year-old female patient was known for mitral valve prolapse since infancy. Her baseline ECG showed negative T waves in the infero-lateral leads (Figure 2A) and mitral regurgitation during echocardiography was moderate (Figure 2B). She was symptomatic for mild palpitations. However, several “red flags” were present (female sex and baseline ECG), as well as a bileaflet prolapse with MAD (Figure 2C) and a Pickelhaube sign (Figure 2D) during transthoracic echocardiography. Considering the patient’s symptoms and the presence of “red flags” in the baseline assessment, she underwent a 24 ECG Holter examination, which showed the presence of several ventricular extrasystoles (>20,000), and the CMR study confirmed the presence of MAD (Figure 2E). In addition, a large zone of fibrosis in the basal inferior wall was visible (Figure 2F,G red circle). The extracellular volume was high in all basal left ventricular segments. In this case, with several “red flags,” an invasive electrophysiology study (EPS) study was undertaken to more precisely stratify the arrhythmic risk. The EPS showed the induction of ventricular tachycardia and even ventricular fibrillation. For this reason, a subcutaneous implantable cardioverter defibrillator (ICD) for primary prevention was finally implanted. The patient was also treated with 2.5 mg of bisoprolol, and a reduction in ventricular arrhythmias was documented at follow-up.


***Case 2 (*Figure 3*): Secondary prevention ICD implantation after cardiac arrest and multi-modality imaging work-up***


A 40 year-old man was known for mitral valve prolapse since adolescence and was periodically followed by a cardiologist. Baseline ECG was not remarkable (Figure 3A), mitral regurgitation was moderate during echocardiography (Figure 3B), and the patient was asymptomatic in his daily life. However, he presented several “red flags” during echocardiography, in particular, the presence of a bileaflet mitral valve prolapse with MAD (Figure 3C). Then, the patient suffered a cardiac arrest on ventricular fibrillation during jogging and he was admitted to the emergency room. Coronary angiography was normal. Moderate mitral regurgitation was confirmed upon urgent echocardiography. During continuous ECG monitoring in the intensive care unit, several ventricular extrasystoles were noticed. CMR was then performed, confirming the presence of MAD (Figure 3D) and demonstrating a large zone of fibrosis in the basal inferior wall (Figure 3E,F, red circle). The extracellular volume determined by T_1_ mapping was high in all basal left ventricular segments. The patient then underwent an EPS, which showed the induction of ventricular tachycardia and ventricular fibrillation. A subcutaneous ICD was finally implanted as a secondary prevention measure, and treatment with 80 mg/day of nadolol was started. No arrhythmic events were present at the one-year follow-up.

## Figures and Tables

**Figure 1 diagnostics-11-00683-f001:**
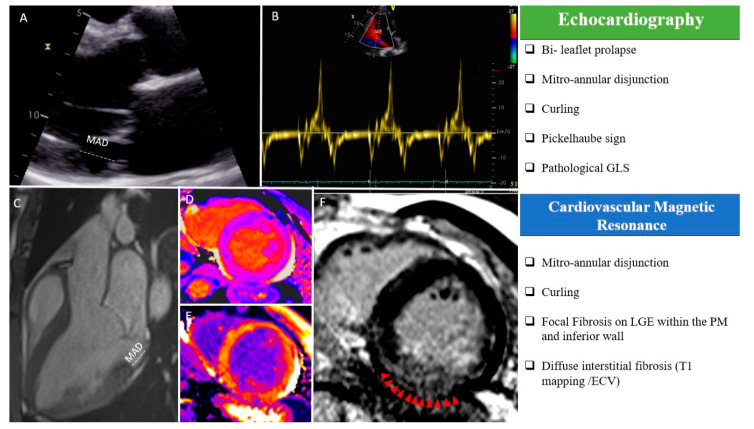
“Red Flags” in echocardiography and cardiac magnetic resonance imaging (CMR): (**A**) MAD measured in a parasternal long-axis view (transthoracic echocardiography); (**B**) “Pickelhaube sign” during transthoracic echocardiography; (**C**) MAD measured in a steady-state free precession three-chamber view during CMR; T1 mapping native (**D**) and after gadolinium injection (**E**); macroscopic fibrosis in the inferior basal wall highlighted with red arrows in a free-breathing LGE sequence (**F**). ECV, extracellular volume; LGE, late gadolinium enhancement; MAD, mitro-annular disjunction; PM, papillary muscle, GLS: Global Longitudinal Strain.

**Figure 2 diagnostics-11-00683-f002:**
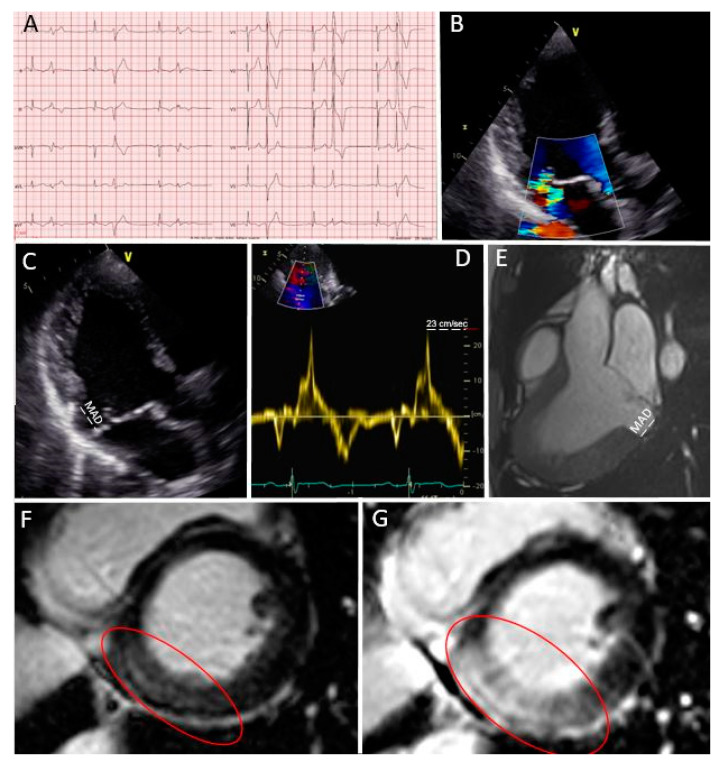
Case 1: Primary prevention ICD implantation after multi-modality imaging work-up. (**A**) Patient’s ECG; (**B**) moderate mitral regurgitation during color Doppler transthoracic echocardiography; (**C**) bileaflet prolapse with MAD (in white); (**D**) “Pickelhaube sign” during transthoracic echocardiography; (**E**) MAD measured in a steady-state free precession three-chamber view during CMR; (**F**,**G**) a large zone of fibrosis in the basal inferior wall in LGE sequences (red circle).

**Figure 3 diagnostics-11-00683-f003:**
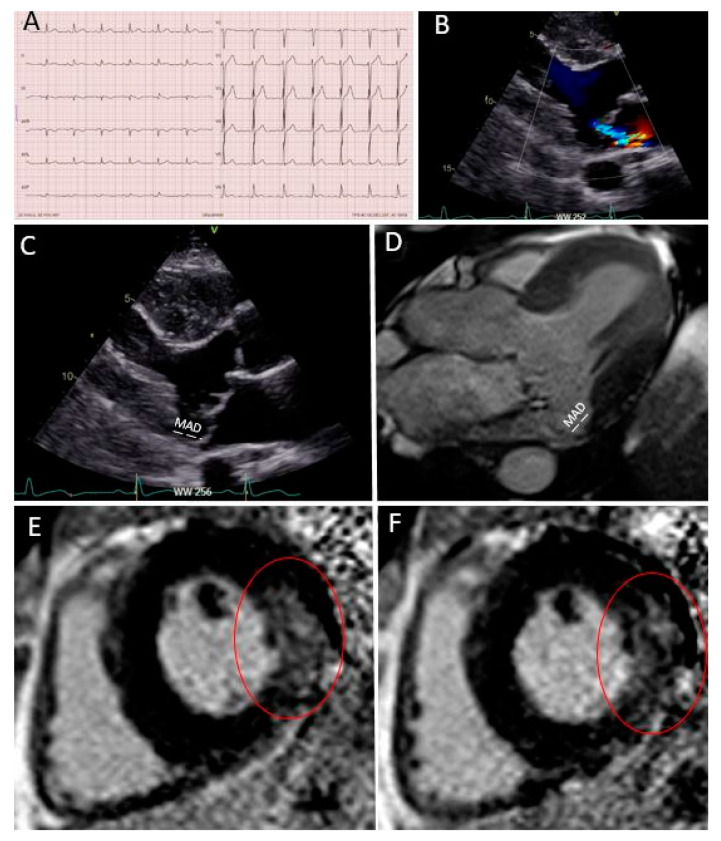
Case 2: Secondary prevention ICD implantation after cardiac arrest and multi-modality imaging work-up. (**A**) Patient’s ECG; (**B**) moderate mitral regurgitation during color Doppler transthoracic echocardiography; (**C**) bileaflet prolapse with MAD (in white); (**D**) MAD measured in a steady-state free precession three-chamber view during CMR; (**E**,**F**) a large zone of fibrosis in the basal and mid-lateral wall in LGE sequences (red circle).

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
