# Peer review of "Mitral Valve Prolapse, Arrhythmias, and Sudden Cardiac Death: The Role of Multimodality Imaging to Detect High-Risk Features"

_diagnostics, 2021, doi:10.3390/diagnostics11040683_

Round 1

Reviewer 1 Report

The manuscript deals with extremely clinically  important topic – arrhythmic mitral valve prolapse,which is characterized by complex premature ventricular beats and increased risk of sudden cardiac death. Thus, defining the red flags or  important morphological features for increased arrhythmic risk in pts with mitral valve prolapse is of great clinical relevance.

The manuscript presents comprehensive  review of high risk features to look for, using up to date imaging techniques, including two clinical cases and management considerations. The authors discuss very thoroughly all the key features of arrhythmic mitral valve prolapse, diagnosed using echocardiography, including tissue Doppler and speckle tracking echocardiography , as well as, CMR with LGE acquisition and T1 mapping. It should be appreciable if the authors expand a little bit  a section concerning inflammation topic.

Very important part of the manuscript is management considerations. This part is presented very clearly and consistently. However,it should be appreciable if the authors explaine more thoroughly the role of T1 mapping in defining the increased arrhythmic risk  and recommend or not recommend T1 mapping in pts with mitral valve prolapse (more  for scientific or  clinical interest?).

The English level of the manuscript is high. The manuscript will be interesting to the wide auditorium of medical specialists, especially cardiologists,sport medicine specialists,etc.

I highly recommend the manuscript for publication, owing to very clinically relevant topic, consistent summary of key features of arrhythmic mitral valve prolapse and recommendations how to identify high risk patients.

Author Response

We are extremely grateful to the reviewer that with his/her comments helps us improving the quality of the manuscript.

We thank the reviewer on his/her interest on the subject of inflammation, which we also find very intriguing. However, it must be considered that since now only an article has been published on the subject and it is still a speculative issue. Anyway, we expanded the section regarding inflammation topic, going into much details of the study published by Miller et. all.

Regarding the use of T1 mapping and ECV we also add a more deepen explanation of its rational in lines 154-157 and in lines 172-176. It is well known that myocardial fibrosis detected by LGE has been associated with the occurrence of ventricular arrhythmia in several ischemic and non-ischemic cardiac conditions. On top of this, from ischemic heart disease we know that areas, which are a higher risk, are characterised by intermingling of both viable myocardium and interstitial fibrosis and may promote re-entry pathways for ventricular arrhythmias. For the moment, no precise guidelines exists on the role of T1 mapping in patients with mitral valve prolapse. Even if for the moment the clinical role is still under evaluation, we strongly recommended its usage in the evaluation of patients with MVP.

Reviewer 2 Report

I have read your article, and it is well-written. This is an interesting topic. Good review with actual case reports. 

  1. There are some misspelling.
  2. Please indicate Pickelhaube sign.
  3. Do you have any data on course in terms of arrythmia if patients have mitral valve surgery?

Author Response

We thank the reviewer for his/her kind comments. We corrected some misspelling errors throughout the manuscript and indicated the Pickelhaube sign.

Regarding patients after mitral valve surgery, we completely agree with the reviewer that it is a very important topic. However, since now, only a small retrospective series have been published on the subject showing that mitral valve surgery apparently does not reduce the burden of ventricular arryhtmias in patients with bileaflets MVP. This study has several limitations and a very small sample size (32 patients) even if it covers 20 years of clinical activities, so at the moment no statistically significant data with a are available on the follow-up in terms of arryhtmias. In any case, we may speculate that since evidence already published on several pathologies highlights the role of replacement fibrosis in the genesis of arrhythmias, if a scar is already present, the arrhythmic burden in those patients may still be an issue even after mitral surgery. However, a multicentric registry is needed to answer this difficult, but interesting question. We update the reviewer in line 222-232 on this interesting topic.